# SLP-Improved DDPG Path-Planning Algorithm for Mobile Robot in Large-Scale Dynamic Environment

**DOI:** 10.3390/s23073521

**Published:** 2023-03-28

**Authors:** Yinliang Chen, Liang Liang

**Affiliations:** 1School of Computer Science, Wuhan University, Wuhan 430072, China; 2School of Power and Mechanical Engineering, Wuhan University, Wuhan 430072, China

**Keywords:** deep reinforcement learning, path planning, mobile robot, deep neural network

## Abstract

Navigating robots through large-scale environments while avoiding dynamic obstacles is a crucial challenge in robotics. This study proposes an improved deep deterministic policy gradient (DDPG) path planning algorithm incorporating sequential linear path planning (SLP) to address this challenge. This research aims to enhance the stability and efficiency of traditional DDPG algorithms by utilizing the strengths of SLP and achieving a better balance between stability and real-time performance. Our algorithm generates a series of sub-goals using SLP, based on a quick calculation of the robot’s driving path, and then uses DDPG to follow these sub-goals for path planning. The experimental results demonstrate that the proposed SLP-enhanced DDPG path planning algorithm outperforms traditional DDPG algorithms by effectively navigating the robot through large-scale dynamic environments while avoiding obstacles. Specifically, the proposed algorithm improves the success rate by 12.33% compared to the traditional DDPG algorithm and 29.67% compared to the A*+DDPG algorithm in navigating the robot to the goal while avoiding obstacles.

## 1. Introduction

In recent years, the prevalence of indoor robotics has significantly increased. These robots are usually assigned to perform different indoor operations, including but not limited to transporting objects, offering navigation assistance, and handling emergencies. To accomplish these tasks effectively, the robot must be able to navigate autonomously from one location to another, making the path planning ability a critical factor in the autonomous behavior of the robot. Moreover, in real-world scenarios, dynamic obstacles, such as human beings or other robots, frequently arise, requiring the robot to also possess the ability to dynamically avoid these obstacles in order to achieve a higher level of autonomy. Therefore, a considerable amount of research has focused on studying path planning and dynamic navigation of robots in indoor environments [1,2,3,4].

Over the past few years, many algorithms have been developed to tackle the challenge of path planning. Among these approaches are the traditional methods, including the genetic algorithm, A* algorithm, D* algorithm, and Dijkstra algorithm, as documented by Gong et al. [5] in their study on efficient path planning. These traditional approaches have demonstrated a degree of efficacy and success. However, some significant limitations have also been identified, such as being time-consuming, having a tendency to converge to local optima, failing to account for collision and risk factors, and so on. Consequently, these prior methods require improvement to meet the demands of more advanced path planning and increasingly complex environments.

Various advanced methods have been proposed in addition to the traditional path planning techniques to improve the path planning process. These include the method proposed by Haj et al. [6], which utilizes machine learning to classify static and dynamic obstacles and applies different strategies accordingly. Bakdi et al. [7] have proposed an optimal approach to path planning, while Zhang et al. [8] have proposed a Spacetime-A* algorithm for indoor path planning. Palacz et al. [9] have proposed BIM/IFC-based graph models for indoor robot navigation, and Sun et al. [10] have employed semantic information for path planning. Additionally, Zhang et al. [11] have proposed a deep learning-based method for classifying dynamic and static obstacles and adjusting the trajectory accordingly. Dai et al. [12] have proposed a method that uses a novel whale optimization algorithm incorporating potential field factors to enhance mobile robots’ dynamic obstacle avoidance ability. Miao et al. [13] have proposed an adaptive ant colony algorithm for faster and more stable path planning, and Zhong et al. [14] have proposed a hybrid approach that combines the A* algorithm with the adaptive window approach for path planning in large-scale dynamic environments. Despite these advances, certain limitations still exist when applying these methods in real-life situations. For instance, Zhang et al.’s model relies on a waiting rule for dynamic obstacles, which may only be suitable for some types of dynamic obstacles. Furthermore, Miao et al.’s adaptive ant colony algorithm does not consider dynamic obstacles, making the model unsuitable for complex indoor environments.

The recent advancements in reinforcement learning have provided novel approaches [15,16,17,18] to address the path planning problem. The success demonstrated by Google DeepMind in various tasks, such as chess, highlights the potential of artificial intelligence in decision-making issues, including path planning. Implementing reinforcement learning in path planning enhances the autonomous capabilities of robots, resulting in safer, faster, and more rational obstacle avoidance. The convergence of the model is expected to provide more optimal solutions in the future. There have been numerous studies that propose reinforcement learning for path planning, such as the SAC model proposed by Chen et al. [19] for real-time path planning and dynamic obstacle avoidance in robots, the DQN model proposed by Quan et al. [20] for robot navigation, and the RMDDQN model proposed by Huang et al. [21] for path planning in unknown dynamic environments.

The sequential linear path (SLP) algorithm was introduced by FAREH et al. [22] in 2020. It is a path planning strategy that emphasizes the examination of critical regions, such as those surrounding obstacles and the goal point. It prioritizes the utilization of processing resources in these areas. As a global path planning approach, SLP generates a comprehensive path from the starting point to the goal while considering static obstacles, breaking down the dichotomy between path quality and speed. In our proposed path planning method, we incorporate the SLP algorithm to enhance the performance of DDPG.

This study presents three key contributions: (1) An enhanced reinforcement learning model for path planning is proposed, which enables the robot to modify its destination dynamically and provide real-time path planning solutions. (2) Both the convergence rate and path planning performance of DDPG are improved by incorporating the use of SLP to generate a global path to guide DDPG in large-scale environment path planning tasks. (3) To validate our approach, we conduct a comprehensive set of simulations and experiments that compare various path planning methods and provide evidence for the effectiveness of our proposed method.

The paper is structured as follows: In Section 2, the problem and related algorithms are described. Section 3 introduces the path planning pipeline of the proposed SLP improved DDPG algorithm, along with a detailed account of the reward function design. Section 4 presents the simulation experiment and a detailed comparison and analysis of the experimental results. Finally, Section 5 offers a comprehensive summary of the paper’s contents.

## 2. Related Work

### 2.1. Problem Definition

As discussed, previous researchers have developed robust and efficient global path planning algorithms. However, the assumption that the path planning environment is static is only valid in ideal scenarios. In reality, most environments for robot path planning, such as factories, warehouses, and restaurants, are dynamic. Thus, relying solely on global path planning will likely result in unsuccessful routing outcomes, as global planning cannot adapt to dynamic obstacles. Robots following a global path in dynamic environments are susceptible to collision with dynamic obstacles, as shown in Figure 1.

### 2.2. Deep Deterministic Policy Gradient

The deep deterministic policy gradient (DDPG) reinforcement learning algorithm was introduced by the Google DeepMind group [23]. It addresses the parameter relevance issue in the Actor–Critic algorithm and the inadequacy of continuous actions in the AQN algorithm. The DDPG algorithm can be considered an improved version of AQN that incorporates the overall structure of the Actor–Critic algorithm. Like the Actor–Critic algorithm, the DDPG algorithm consists of two primary networks: the Actor and Critic networks.

The output of the Actor network is a determined action, which is defined as $a=\mu (s,a|\theta^{\mu })$. The estimation network that produces real-time actions is the $\mu_{\theta }\left(s\right)$, where we have $\theta^{\mu }$ indicating the parameters. It updates the parameter $\theta^{\mu }$, outputs the action A according to the current state $s_{t}$ and interacts with the environment to generate the next state $s_{t+1}$ and reward $r_{t+1}$. The Actor target network updates parameters in the Critic network and determines the following optimal action $a_{t+1}$ according to the next state $s_{t+1}$ sampled from the experience replay.

The Critic network aims to fit the value function $Q(s,a|\theta^{Q})$. The Critic estimation network updates its parameters $\theta^{Q}$, calculates the current Q value $Q(s_{t},a_{t},\theta^{Q})$ and the target Q value $y_{i}=r+\gamma Q^{\prime }\left(s_{t+1},a_{t+1},\theta^{Q^{\prime }}\right)$. The Critic target network calculates the $Q^{\prime }$ part of the target Q value. Here, γ is the discount factor that has the range γ∈[0,1].

The training process of the Critic network is to minimize the loss function L shown in Equation (Equation 1).
(1)$$L=\frac{1}{N}\sum_{i}{\left(y_{i}-Q\left(s_{i},a_{i}|\theta^{Q}\right)\right)}^{2}$$
where $y_{i}$ is given by Equation (Equation 2).
(2)$$y_{i}=r\left(s_{i},a_{i}\right)+\gamma Q^{\prime }\left(s_{i+1},\mu^{\prime }\left(s_{i+1}|\theta^{\mu^{\prime }}\right)|\theta^{Q^{\prime }}\right)$$

The Actor network is updated through a policy gradient shown in Equation (Equation 3).
(3)$$\nabla_{\theta^{\mu }}J\approx \frac{1}{N}\sum_{i}\nabla_{a}Q\left(s,a\;|\theta^{Q}\right){{|}_{s=s_{i},a=\mu \left(s_{i}\right)}\nabla_{\theta^{\mu }}\mu \left(s|\theta^{\mu }\right)|}_{s_{i}}$$

Target networks are updated through Equation (Equation 4), with τ≪1.
(4)$$\begin{array}{cc} \theta^{Q^{\prime }} & \leftarrow \tau \theta^{Q}+\left(1-\tau \right)\theta^{Q^{\prime }}\\ \theta^{\mu^{\prime }} & \leftarrow \tau \theta^{\mu }+\left(1-\tau \right)\theta^{\mu^{\prime }} \end{array}$$

Comparing DDPG with previous path planning and dynamic obstacle avoidance methods, such as ant colony, A* + adaptive window, roaming trail, etc., we may find DDPG has more potential for high complexity dynamic environment path planning and dynamic obstacle avoidance due to the following reasons:1.DDPG is a reinforcement learning method that improves through multiple iterations and gradually converges, which will likely provide optimal solutions for the current environment and state.2.Unlike traditional path planning methods that require previous knowledge of the environment map, DDPG does not require any map but can reach the goal using a pre-trained model. This makes the DDPG method more adaptive since, in dynamic environments, it is hardly possible to track the environment’s moving obstacles’ trajectory.3.Unlike traditional means of dynamic obstacle avoidance that produce simple actions when the robot encounters a dynamic obstacle, DDPG outputs a series of actions. The actions that DDPG produces to avoid obstacles are relatively reasonable and may have connections with previous actions. It is a real-time obstacle avoidance. The actions may be planned ahead of time instead of stopping the robot from adjusting the trajectory when a dynamic obstacle is detected.

DDPG presents excellent potential in performing small-scale path planning. However, DDPG is a reinforcement learning method, which inevitably inherited the characteristics of randomness and uninterpretability of RL, thus causing the path planning results of DDPG to represent signs of randomness and unreasonableness. This is one of the defects of DDPG in path planning since it could cause inefficiency and is exaggerated as the scale of the environment increases. An example of DDPG path planning failing the path planning task or being unable to perform the task efficiently in a large-scale environment is shown in Figure 2.

### 2.3. Sequential Linear Paths

The SLP (sequential linear paths) approach is a path planning method put forward by Fareh et al. in 2020 that has overcome the traditional trade-off between execution speed and path quality (also known as the swiftness-quality problem), enhancing both the path quality and swiftness.

The SLP path planning strategy focuses more on areas around static obstacles and areas around the goal point, defined as critical areas. SLP exhausts the processing power in critical areas to reduce computational resource waste and improve swiftness. It is interested in the obstacles that intersect with the direct linear line between the starting and goal points while ignoring unnecessary obstacles that do not lie alongside the route.

The SLP is a hybrid approach that implements traditional path planning, such as A*, D*, or the probabilistic roadmap (PRM), to find possible paths around obstacles that intersect with the linear line from the robot to the destination. The implemented procedure may reduce the computational efforts compared to applying the basis algorithm on the whole map. SLP enhances the path quality by finding any linear shortcuts between any two points in the path and modifying these shortcuts as the final global path.

The experimental results have proven that the SLP path planning strategy showed a superior advantage over other path planning techniques in both aspects, computational time (reached up to 97.05% improvement) and path quality (reached up to 16.21% improvement for path length and 98.50% for smoothness).

Our path planning method used SLP to calculate a prior global route to guide the DDPG to improve its performance in large-scale environments. The main idea is to perform global path planning using SLP and select sub-goal points to provide directional aid for the DDPG.

The sub-goals SG between the starting point *S* and goal point *G* are directly generated using the SLP algorithm, and thus, the path can be denoted using points:(5)$$Path=\{S,SG_{1},SG_{2},\cdots ,G\}$$

## 3. SLP Improved DDPG Path Planning

### 3.1. Path Planning Based on Improved DDPG Algorithm

When considering the application of reinforcement learning in small to medium-scale dynamic environments for path planning, the model is likely to converge and achieve satisfactory results after sufficient training. However, in large-scale dynamic environments, where the distance between the starting point and the goal may be substantial, the model may exhibit a low convergence rate and poor test results. The static path length (SLP) algorithm exhibits exceptional performance in static environments for path planning, but it may not be suitable for dynamic environment path planning. Conversely, the reinforcement learning method deep deterministic policy gradients (DDPG) demonstrates competency in dynamic obstacle avoidance through rational and continuous decision-making. Thus, SLP is an excellent static environment path planning method, while DDPG is well-suited for dynamic environments. To address the challenge of reinforcement learning for large-scale path planning, we propose an improved DDPG path planning algorithm incorporating the SLP method to leverage the strengths of both approaches.

Upon closer examination of the issue, it becomes evident that the key challenge lies in that, in large environments, the RL model may require a significant number of iterations to reach the goal, resulting in a low convergence rate and reducing its accuracy and robustness. Given that the RL model is responsible for local path planning, training the model in small-scale environments and using SLP to enhance the pre-trained model is recommended. The proposed path planning pipeline involves the following steps:

The first step in the process is to obtain the environment grid map by having the robot navigate the area and collect sensor data. Next, a path and sub-goals are generated using the SLP algorithm based on the collected map, considering only static obstacles. The robot then attempts to reach the sub-goals using a pre-trained DDPG model, with the expectation that the model will avoid obstacles along the way. Generally, the SLP algorithm is anticipated to provide the primary path, while the DDPG model handles obstacle avoidance. Figure 3 depicts the proposed path planning pipeline, and the pseudo-code is presented in Algorithm 1.
**Algorithm 1** SLP-improved DDPG path planning algorithm**Input:** 
Map: Environment static map.**Input:** 
S,G: Start and goal’s coordinates.**Input:** 
x,y: Current self coordinates.**Input:** 
$d_{thresh}$: Method switch distance threshold.**Input:** 
*D*: LiDAR sensor’s distances.**Input:** 
ϵ: Goal threshold.**Input:** 
Dynamic: Pre-trained DDPG model.**Input:** 
MAXTIME: Timeout threshold.**Ensure:** 
success_state      success_state←False,length←0      Path←SLP(S,G,Map)// Apply the SLP method to generate a global path      Path←SelectSubGoals(Path)// Select sub-goals      **for** 
$SG=SG_{1},SG_{2}\cdots ,SG_{n},G\in Path$
 **do**           **while** $\sqrt{{\left(x-SG.x\right)}^{2}+{\left(y-SG.y\right)}^{2}}>\epsilon$ **do**             a←Dynamic(D,x,y,SG)             **if** $TimeElapsed>MAXTIME\;\mathrm{or}\;\exists d_{i}\in D\;d_{i}\leq 0.13$ **then**                 **return** success_state //Fail when timeout or collision              **end if**           **end while**      **end for**      success_state←True      **return**
success_state

### 3.2. DDPG Reward Function Design

We use π to denote the policy specifying the action set the agent should take given state $s_{t}$. In Equation (Equation 6), *S* denotes the set of possible states, $A(s_{t})$ is the action set given state $s_{t}$, and $a_{t}$ and $s_{t}$ are the action and state elements in action space $A(s_{t})$ and state space *S*.
(6)$$\pi :a_{t}\in A\left(s_{t}\right),\;s_{t}\in S$$

Next, we designed a reward principle based on state $s_{t}$ and action $a_{t}$ shown in Equation (Equation 7). Here $d_{i}$ denotes the distance data detected by the i th LiDAR sensor, and D is the matrix of the robot’s LiDAR scan ranges shown in Equation (Equation 8). Aside from direct rewards for goal and collision, there are two other rewards $r_{\mathrm{obs}}$ and $r_{\mathrm{yaw}}$. $r_{\mathrm{obs}}$ is the obstacle avoidance reward, shown in Equation (Equation 9). Here $d_{\min}$ is the minimum LiDAR scan range in matrix *D*. $r_{\mathrm{yaw}}$ is the action select reward, shown in Equation (Equation 9). Here *h* is the angle between the robot’s heading and the goal, and k(k=0,1,2,3,4) is the action serial number indicating actions of turn left, turn front left, front, turn front right and turn right.
(7)$$r\left(s_{t},a_{t}\right)=\left\{\begin{array}{cc} r_{\mathrm{goal}} & s_{t}=s_{\mathrm{goal}}\\ r_{\mathrm{collision}} & s_{t}=s_{\mathrm{collision}}\\ r_{\mathrm{obs}}+r_{\mathrm{yaw}} & otherwise \end{array}\right.$$
(8)$$D=[d_{1},d_{2},\cdots ,d_{n}]$$
(9)$$r_{obs}=\left\{\begin{array}{cc} -\min(2^{\frac{1}{d_{\min}}},50) & d_{\min}<0.7\\ 0 & otherwise \end{array}\right.$$
(10)$$r_{\mathrm{yaw}}=1-4|\frac{1}{2}-\left(\frac{1}{4}+\frac{1}{2\pi }\left(\left(h+\frac{\pi }{8}k+\frac{\pi }{4}\right)\;\mathrm{mod}\;2\pi )\right)\right|$$

State $s_{\mathrm{goal}}$ and $s_{\mathrm{collision}}$ are defined in Equations (Equation 11) and (Equation 12), respectively.
(11)$$s_{\mathrm{goal}}:\sqrt{{\left(x-x_{\mathrm{goal}}\right)}^{2}+{\left(y-y_{\mathrm{goal}}\right)}^{2}}\leq 0.1$$
(12)$$s_{\mathrm{collision}}:\exists d_{i}\in D\;d_{i}\leq 0.13\;$$

## 4. Experiments and Results

To assess the effectiveness of our methodology, we developed a simulation environment utilizing Gazebo. The Turtlebot3 platform implemented our approach within this environment, including static and dynamic obstacles to simulate realistic conditions. We evaluated the performance of our approach using three fundamental metrics: average execution time (AET), average path length (APL), and success rate (SR). Notably, a higher success rate can lead to longer execution times and path lengths, while early collisions may result in shorter path lengths and execution times. To provide a normalized assessment, we integrated two additional metrics, time index (TI) and path length index (PLI), to offer a relative performance measure. The equations for computing these indices are presented below:(13)$$TI=\frac{AET}{SR}$$
(14)$$PLI=\frac{APL}{SR}$$

### 4.1. Experimental Setup

In this study, the experiment was conducted on an Intel Core i9 CPU(3.5 GHz) with 32 GB of RAM running the Ubuntu Xenial Xerus distribution of Linux. Python 2.7 and PyTorch 1.0.0 were used to realize the proposed algorithm. The simulation environment was established using ROS 1.12.17 and Gazebo7, with the Turtlebot3 burger robot as the model. The DDPG algorithm was trained over 3000 episodes within the environment to generate a pre-trained reinforcement learning model, and the average cumulative reward was computed. As shown in Figure 4, during the initial stages of training, the robot was in the exploration and learning phase, resulting in a relatively low average cumulative reward. However, as the number of episodes increased, the robot began utilizing its experiences, causing the number of successful target reaches to increase and the average cumulative reward to rise accordingly. After 1500 episodes, the model converged, resulting in a reward of approximately 4000 at its conclusion. To evaluate the validity of the proposed path planning model, its performance was compared against well-established path planning methods, such as A*, SLP, DDPG, and the A*-improved DDPG method.

### 4.2. Grid Map Generation

The limitations of DDPG can be mitigated and improved by utilizing a global route that divides the large-scale environment path planning tasks into smaller-scale path planning tasks.

Since SLP is a global path planning algorithm, obtaining the grid map is a prerequisite for SLP to generate a global path. To acquire the grid map, we utilize a robot equipped with LiDAR to manually scan the environment and reconstruct the raw grid map through SLAM using the collected LiDAR data. An illustration of the raw grid map can be seen in Figure 5b.

However, using the raw grid map for global path planning leads to inevitable failures as the global planning algorithms must incorporate the robot’s collision volume. Modifying the SLP planning algorithm to account for the collision volume increases computation time, which is burdensome. An alternative approach is to add safe padding to the raw grid map. This allows the SLP to consider both the obstacles and the padding as obstacles during global path generation, thereby mitigating the likelihood of collisions with static obstacles while following the global path to some extent.

The raw grid map is stored in a two-dimensional array, transforming the spatial correlation to an array index correlation. We use an array element value to denote obstacles (0 for non-obstacle and 1 for obstacle). The grid map padding is added through Equation (Equation 15), where *O* is the set of obstacles in raw grid map, $x_{ij}$ is the element in a padded grid map with index i,j, $o_{mn}$ is the obstacle element in raw grid map with index m,n and thresh is an arbitrary threshold. An example of a padded grid map is shown in Figure 5c.
(15)$$\begin{array}{cc} d(x_{ij},o_{mn}) & =\sqrt{{\left(i-m\right)}^{2}+{\left(j-n\right)}^{2}}\\ x_{ij} & =\left\{\begin{array}{cc} 1 & \;\mathrm{if}\;\exists o_{mn}\in O\;\mathrm{that}\;d\left(x_{ij},o_{mn}\right)<thresh\\ 0 & \;\mathrm{otherwise}\; \end{array}\right. \end{array}$$

After generating the grid map, we now possess a padded grid map for using SLP to generate a global path and conduct a simulation.

### 4.3. Simulation Results in Static Environments

Our initial evaluation was conducted in a static environment, the configuration of which is depicted in Figure 6. The target location was randomly selected within the constraints of the environment, while the robot was reset to a fixed starting point. To perform path planning and obstacle avoidance in the static environment, we employed the following methods: A*, SLP, DDPG, Hybrid Method A* Improved DDPG, and our Proposed Method.

The experimental data are recorded in Table 1, the robot trajectory comparisons are presented in Figure 7, and the trajectories of different methods are illustrated in Figure 8. The results indicate that the SLP method has a 30.5% reduction in path length compared to the A* method, albeit with a higher average execution time and lower success rate. However, these limitations of SLP can be addressed through integration with DDPG. The results suggest that the SLP-enhanced DDPG method has reduced the average execution time by 46.58% and increased the success rate by 52.67% compared to SLP alone. The SLP method substantially improves path length compared to the traditional A* method. However, its lower success rate may stem from its characteristic of planning a path around obstacles, which increases the likelihood of a collision. The DDPG reinforcement learning method exhibits a high success rate of 94% in obstacle avoidance. The combination of SLP and DDPG yields a method with a relatively short path length and high success rate. The results in Table 1 demonstrate that the overall performance of the SLP-enhanced DDPG method in static environments is superior to the other four algorithms.

### 4.4. Simulation Results in Dynamic Environments

Subsequently, the efficacy of our proposed path planning technique is evaluated in a dynamic environment where two randomly moving cylindrical obstacles are present. The schematic representation of this environment is depicted in Figure 9. We employed the A* Algorithm, SLP, DDPG, the Hybrid Method incorporating A* and Improved DDPG, and our proposed technique to carry out path planning and obstacle avoidance in this environment.

The experimental data are recorded in Table 2, the robot trajectory comparisons are presented in Figure 10, and the trajectories of different methods are illustrated in Figure 11. A comparison between the success rate and previous static data reveals a decrease in the success rate of all algorithms by a certain percentage, likely due to the dynamic nature of the environment, thus highlighting the challenge of path planning in dynamic environments. The DDPG method demonstrated the highest success rate of 87.67%, indicating its ability to avoid obstacles. Our proposed method’s TI, PLI, and success rate were slightly lower than DDPG under a small-scale dynamic experiment environment. The results in Table 2 show that the SLP-improved DDPG method performs similarly to the DDPG method and is superior to the A*, SLP, and A*-improved DDPG algorithm in the dynamic environment.

### 4.5. Simulation Results in Large-Scale Dynamic Environments

Subsequently, the efficacy of our proposed path planning algorithm was evaluated in a large-scale dynamic scenario. As the term “large” is ambiguous, a scenario with six randomly oscillating cylindrical obstacles was selected. The robot speed was 0.15 m/s, and the environment size was nine times greater than that of the previously employed dynamic scenario. This environment is depicted in Figure 12. For path planning and obstacle avoidance, DDPG, a hybrid approach combining A* with DDPG, and our proposed method were employed, while static methods were omitted due to their poor performance in large-scale dynamic environments.

The experiment data, recorded in Table 3, indicates that the success rate of all methods has decreased due to the increased dynamism and complexity of the experiment environment. The results reveal that the proposed SLP-improved DDPG method has the highest success rate, which is 21% higher than the A*-improved DDPG method and 10.33% higher than the DDPG method. This further demonstrates that the proposed method has improved performance in completing path planning tasks in large-scale dynamic environments. However, in this case, the proposed method’s TI and PLI are inferior to DDPG. Table 3 shows that the SLP-improved DDPG method’s obstacle avoidance performance in large-scale dynamic environments is superior to other methods. At the same time, its path quality is inferior to the DDPG method.

In order to establish the robustness and efficacy of our proposed methodology, we undertook additional experiments by varying the velocity of the robotic system and the number of dynamic obstacles. Specifically, we conducted two more specific test cases, Test Case 4 and Test Case 5, with different settings. Test Case 4 incorporated four dynamic obstacles and a robot speed of 0.3 m/s, whereas Test Case 5 featured ten dynamic obstacles with a reduced robot speed of 0.1 m/s. The outcomes of these experiments have been meticulously documented and are presented in Table 4 and Table 5, respectively.

The recorded experimental data presented in Table 4 reveals a reduction in path length and execution time for all methods compared to Test Case 3. This outcome is attributed to the increased speed of the robot and the reduced number of dynamic obstacles. The results expose that the proposed SLP-improved DDPG method boasts the highest success rate, exceeding the A*-improved DDPG method by 29.67% and the DDPG method by 12.33%. These findings demonstrate that the proposed method’s performance in completing path planning tasks in large-scale dynamic environments is enhanced. It is worth noting that, in this case. At the same time, the DDPG method records the lowest AET and APL, and the SLP-improved DDPG method scores the lowest TI and PLI, reducing by 11.72% and 12.10%, respectively, suggesting improved path planning efficiency and path quality. Table 4 shows that the SLP-improved DDPG method outperforms other methods concerning obstacle avoidance performance and path quality in large-scale dynamic environments.

The experimental data presented in Table 5 demonstrates a clear trend in the increase in both path length and execution time and a decrease in success rate for all methods compared to Test Case 3. This outcome can be attributed to the reduced speed of the robot and an increased number of dynamic obstacles. The results reveal that the proposed SLP-improved DDPG method achieved the highest success rate, exceeding the A*-improved DDPG method by 21% and the DDPG method by 2%. These findings demonstrate that the proposed method outperforms other methods in completing path planning tasks in large-scale dynamic environments. Notably, the SLP-improved DDPG method also exhibits the lowest average execution time (AET), average path length (APL), time inefficiency (TI), and path length inefficiency (PLI). Compared to the DDPG method, the SLP-improved DDPG method reduced TI by 7.62% and PLI by 7.05%, suggesting improved path planning efficiency and path quality. Furthermore, the results presented in Table 4 confirm that the SLP-improved DDPG method surpasses other methods regarding obstacle avoidance performance and path quality in large-scale dynamic environments. These findings underscore the effectiveness of the proposed SLP-improved DDPG method for addressing the challenges of path planning in complex and dynamic environments.

## 5. Conclusions

This study proposes a novel approach for path planning of mobile robots in large-scale environments. The method considers dynamic and static obstacles and prioritizes efficiency and real-time performance in the path planning process. The proposed approach improves the DDPG method and integrates a global planning strategy using the SLP method. The pipeline of the proposed method consists of the following steps: obtaining the environment grid map, executing SLP global planning and generating sub-goals, and finally, using the DDPG algorithm to navigate the environment while avoiding obstacles. The performance of the algorithm was verified through Gazebo simulations. The results of the path planning and collision avoidance tasks in three scenarios demonstrate that the SLP-improved DDPG algorithm has the capability for path planning and dynamic collision avoidance in compliance with robots. Additionally, various other path planning methods were analyzed and compared. The experimental results indicate that the algorithm has the advantages of real-time and safety in avoiding multiple dynamic obstacles, thereby improving the robots’ efficiency in such scenarios.

## Figures and Tables

**Figure 1 sensors-23-03521-f001:**
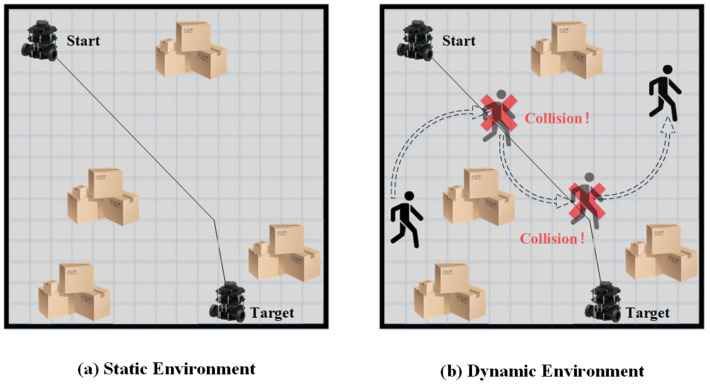
Problem of Global Path Planning in Dynamic Environment.

**Figure 2 sensors-23-03521-f002:**
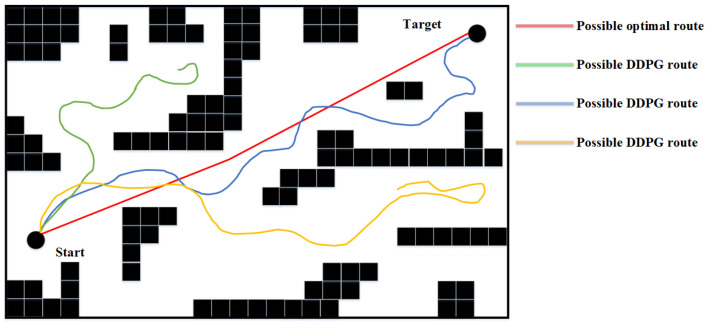
Problem of DDPG Path Planning in Large Environment.

**Figure 3 sensors-23-03521-f003:**
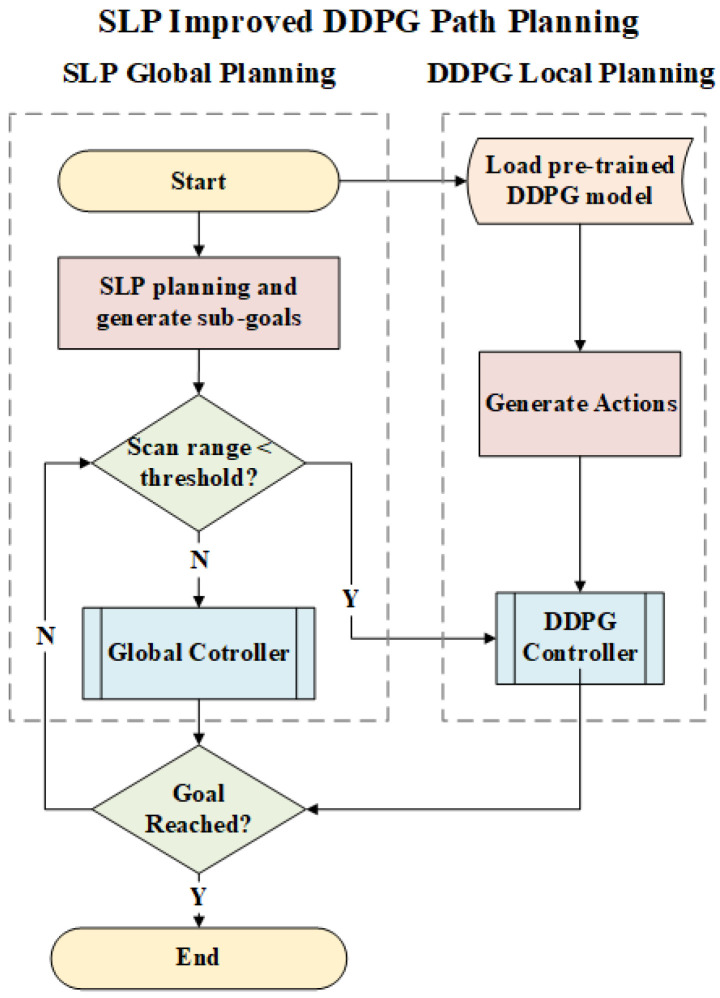
Process Graph.

**Figure 4 sensors-23-03521-f004:**
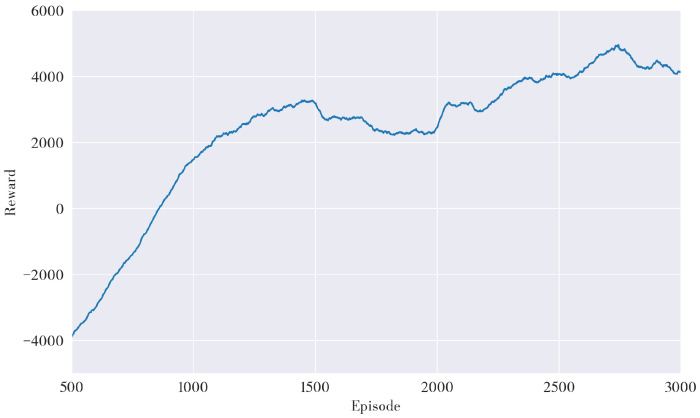
Average Cumulative Reward of DDPG Algorithm.

**Figure 5 sensors-23-03521-f005:**
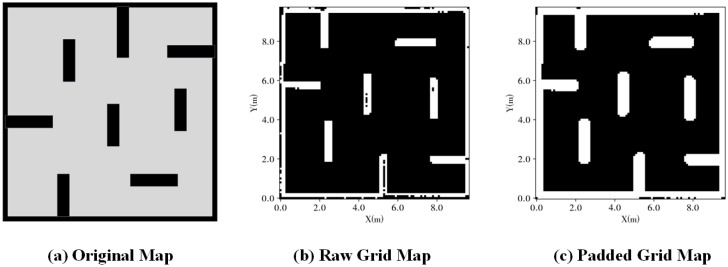
Grid Map Padding.

**Figure 6 sensors-23-03521-f006:**
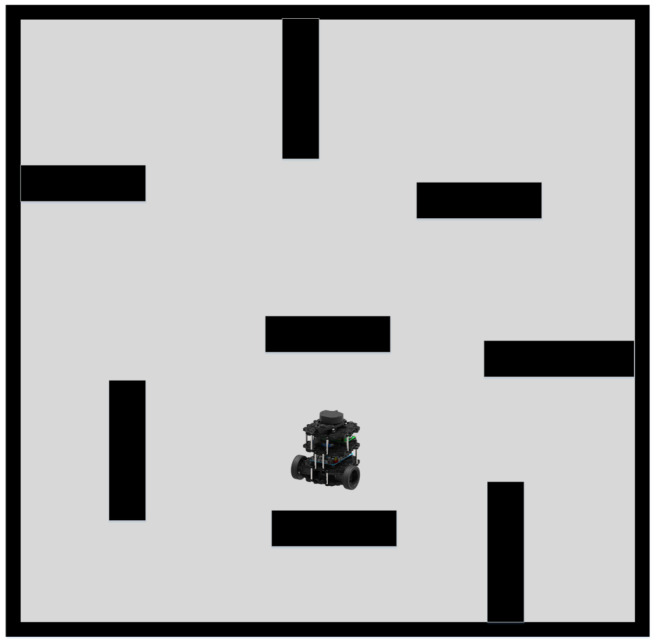
Static Environment.

**Figure 7 sensors-23-03521-f007:**
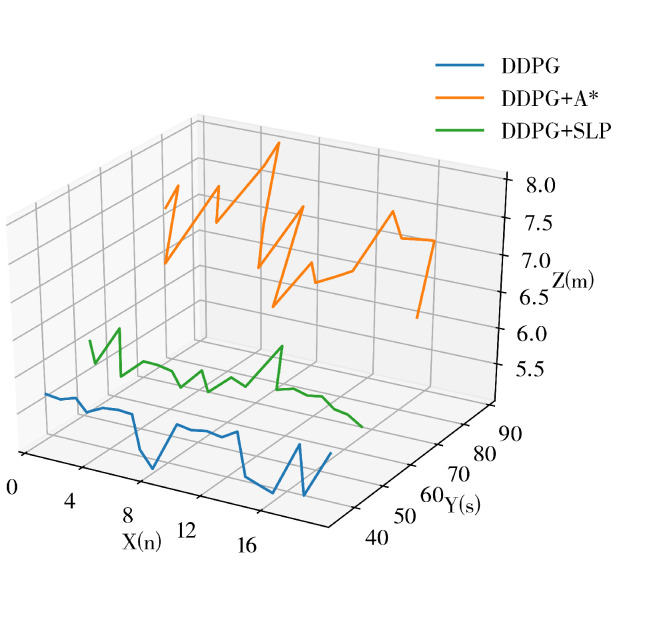
Trajectory Comparison in Static Environment.

**Figure 8 sensors-23-03521-f008:**
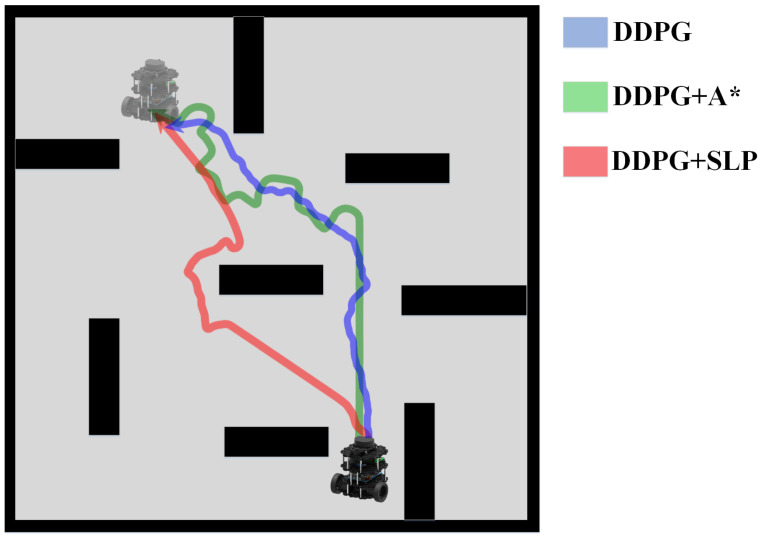
Robot Trajectory of Different Method in Static Environment.

**Figure 9 sensors-23-03521-f009:**
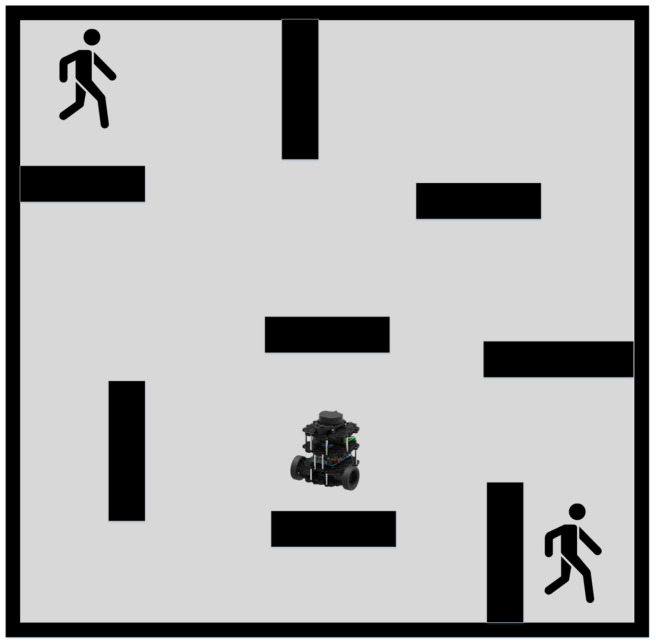
Dynamic Environment.

**Figure 10 sensors-23-03521-f010:**
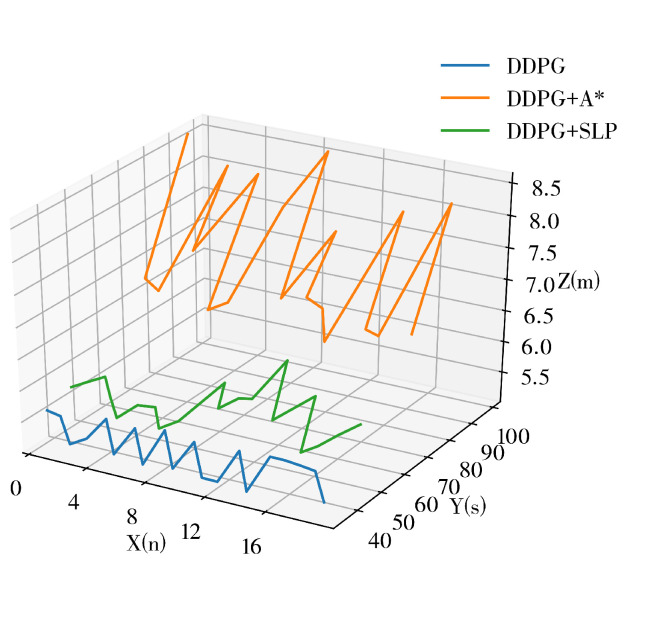
Trajectory Comparison in Dynamic Environment.

**Figure 11 sensors-23-03521-f011:**
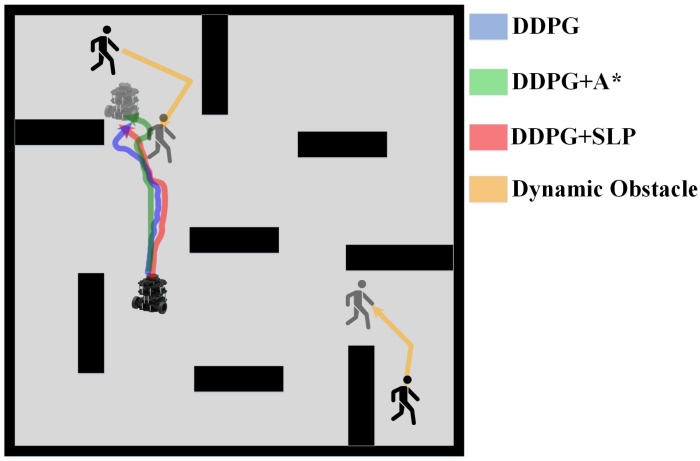
Robot Trajectory of Different Method in Dynamic Environment.

**Figure 12 sensors-23-03521-f012:**
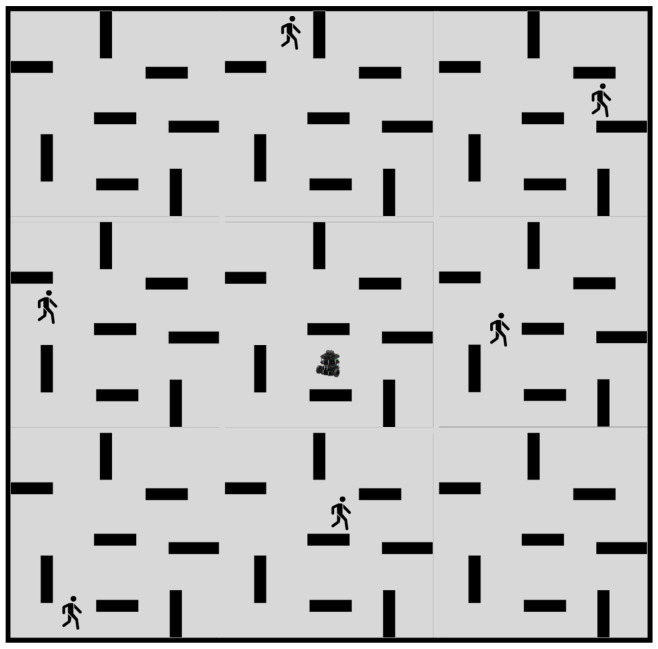
Large-Scale Dynamic Environment.

**Table 1 sensors-23-03521-t001:** Test Case 1 (Static Environment).

Algorithm	AET (sec)	TI	APL	PLI	SR (%)
A*	32.15	32.26	2.92	2.93	99.67
SLP	43.13	67.19	2.03	3.16	64.19
DDPG	19.28	20.51	2.67	2.84	94
A*+DDPG	29.76	38.01	2.66	3.40	78.3
SLP+DDPG	23.04	23.51	2.42	2.47	98

**Table 2 sensors-23-03521-t002:** Test Case 2 (Dynamic Environment).

Algorithm	AET (sec)	TI	APL	PLI	SR (%)
A*	27.84	34.09	2.81	3.44	81.67
SLP	27.67	48.83	2.10	3.71	56.67
DDPG	19.78	21.42	2.75	2.98	92.33
A*+DDPG	25.18	35.63	2.27	3.21	70.67
SLP+DDPG	19.61	22.63	2.64	3.04	86.67

**Table 3 sensors-23-03521-t003:** Test Case 3 (Large-Scale Dynamic Environment).

Algorithm	AET (sec)	TI	APL	PLI	SR (%)
DDPG	43.69	75.33	6.35	10.95	58
A*+DDPG	62.60	132.26	8.88	18.76	47.33
SLP+DDPG	59.18	86.61	8.59	12.57	68.33

**Table 4 sensors-23-03521-t004:** Test Case 4 (Large-Scale Dynamic Environment).

Algorithm	AET (sec)	TI	APL	PLI	SR (%)
DDPG	26.88	48.28	7.68	13.80	55.67
A*+DDPG	34.76	88.38	9.45	24.03	39.33
SLP+DDPG	29.41	42.62	8.37	12.13	69

**Table 5 sensors-23-03521-t005:** Test Case 5 (Large-Scale Dynamic Environment).

Algorithm	AET (sec)	TI	APL	PLI	SR (%)
DDPG	59.93	156.35	5.82	15.18	38.33
A*+DDPG	79.00	408.69	7.72	39.94	19.33
SLP+DDPG	58.25	144.43	5.69	14.11	40.33

## Data Availability

The data used to support the findings of this study are available from the corresponding author upon reasonable request (liangliang@whu.edu.cn).

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
