# Peer review of "SLP-Improved DDPG Path-Planning Algorithm for Mobile Robot in Large-Scale Dynamic Environment"

_sensors, 2023, doi:10.3390/s23073521_

Round 1
Reviewer 1 Report
This manuscript proposes an improved deep deterministic policy gradient (DDPG) path planning algorithm that incorporates sequential linear path planning (SLP) for large-scale environments with dynamic obstacles. On the whole, the research content of this manuscript is relatively meaningful, but there are also some problems that need to be modified.
1. The abstract is an important part of an article. It is suggested that the content of the Abstract should be reorganized, the research purpose of this manuscript should be given at first, that is, why it is necessary to study this manuscript.
2. To facilitate readers to understand and read the manuscript, it is suggested to give the section arrangement of the manuscript in the last paragraph of Introduction.
3. It is suggested to change formula. X to Equation (X) in the manuscript. “formula. X” is not the conventional form of elaboration in academic papers.
4. Units are missing in some Figures, such as Figure 5. Please check other Figures
5. It is suggested to sort out the contents of this manuscript and use less “ in this paper”.
6. The language of the manuscript suggests progressive polishing to improve the quality of the manuscript.
Reviewer 2 Report
The paper proposed a path planning algorithm that integrates SLP global planer and DDPG based local planner. The paper has generally a good clarity and readability. However, the manuscript should be further improved in the following aspects:
1. The abstract may include more specific performance metrics achieved by the proposed algorithm.
2. The literature review was not conducted properly. Some citations in Section 1 were listed in the context without meaningful interpretation.
3. The content of Section 2 was not well organized.
4. Some description of DDPG may be moved to the earlier section, instead of section 3.2.
5. The experimental setting of the simulations should be further clarified.
6. The performance evaluation was also quite confusing without necessary descriptions.
7. The simulation scenario only covers one robot which imposes less challenge. The mobility/speed of the robot and number of the robots/pedestrians in the scene should be considered in the evaluation.
Reviewer 3 Report
The authors present a paper that discusses a SLP improved DDPG path planning algorithm for mobile robot . My comment (mostly editorial) to the authors is that some references are out of date, the author should update the references in the revised manuscript. In my opinion the paper is well written and is overall of interest.
Round 2
Reviewer 1 Report
According to the comments of the reviewer, the authors of manuscript have made detailed revisions to this manuscript, making the manuscript’s content more abundant and reasonable, and the language more fluent. The formula format meets the requirements of Sensors. Therefore, the reviewer believes that the manuscript can be acceptable.
Reviewer 2 Report
The authors have addressed the reviewer's comments carefully and revised the paper accordingly. The quality of the manuscript has been much improved. Acceptance with minor revision is recommended.
I believe double checking of typos in the final version would be fine for the minor revision of this submission.
